# HIV testing, care and viral suppression among men who have sex with men and transgender individuals in Johannesburg, South Africa

**Elizabeth Fearon**[1]\*, **Siyanda Tenza**[2], **Cecilia Mokoena**[2], **Kerushini Moodley**[2], **Adrian D. Smith**[3], **Adam Bourne**[4], **Peter Weatherburn**[5], **Thesla Palanee-Phillips**[2]

**1** Department of Global Health and Development, London School of Hygiene and Tropical Medicine, London, United Kingdom, **2** Wits Reproductive Health and HIV Institute, Johannesburg, South Africa, **3** Nuffield Department of Population Health, University of Oxford, Oxford, United Kingdom, **4** Australian Research Centre in Sex, Health and Society, LaTrobe University, Melbourne, Australia, **5** Department of Public Health, Environments and Society, London School of Hygiene and Tropical Medicine, London, United Kingdom

\* Elizabeth.Fearon@lshtm.ac.uk

**Data Availability Statement:** The data underlying this study cannot be made publicly accessible due to concerns about confidentiality when combining

## Abstract

### Introduction

Men who have sex with men and transgender individuals (MSM/TG) carry a disproportionately high burden of HIV, including in South Africa. However, there are few empirical population-representative estimates of viral suppression and the HIV care cascade including HIV testing among this population, nor of factors associated with these outcomes.

### Methods

We conducted a respondent driven sampling (RDS) survey among 301 MSM/TG in Johannesburg in 2017. Participants gave blood samples for HIV testing and viral load. Participants self-completed a survey including sociodemographics, HIV testing history, and engagement in care. We calculated RDS-II weighted estimates of the percentage of HIV-negative MSM/TG reporting HIV testing in the previous 6 months, their testing experience and preferences. Among those HIV-positive, we estimated the percentage status-aware, on ART, and virally suppressed (<50 viral copies/ml plasma). We conducted RDS-weighted robust Poisson regression to obtain weighted prevalence ratios of factors associated with 1) HIV testing among those HIV-negative; and 2) viral suppression among those HIV-positive.

### Results

There were 118/300 HIV-positive MSM/TG, (37.5%). Of the HIV-negative MSM/TG, 61.5% reported that they had tested for HIV in the previous 6 months, which was associated with selling sex to men (Prevalence Ratio = 1.67, 95% CI 1.36–2.05). There were 76/118 HIV-positive MSM/TG (56.5%) who reported having previously tested positive for HIV and 39/118 (30.0%) who reported current ART. There were 58/118 HIV-positive MSM/TG with viral loads <50 copies/ml plasma (46.9%). Viral suppression was associated with older age

sensitive data about individuals and the respondent driven sampling (RDS) recruitment information needed to reflect the sampling design in analyses. Although we cannot make study data publicly accessible at the time of publication, all authors commit to make the data underlying the findings of the study available in compliance with the PLOS Data Availability Policy. Data can be requested via the London School of Hygiene and Tropical Medicine Research Operations Office Data Management Lead: alex.hollander@lshtm.ac.uk and the first author (elizabeth.fearon@lshtm.ac.uk) and Principal Investigator (tpalanee@wrhi.ac.za). Individuals requesting data should give their research objectives and indicate the list of requested variables, see S3 Survey Instrument. To protect the confidentiality of participants, data sharing is contingent on the data being handled appropriately by the data requester and in accordance with all applicable local requirements.

**Funding:** This study was funded through the Evidence for HIV prevention in Southern Africa (EHPSA) programme (http://www.ehpsa.org/, grant number MM/EHPSA/WHC/0116029, PI Thesla Palanee-Phillips) funded by UK aid from the Department for International Development, and Sweden, through the Swedish International Development Agency (SIDA), mandated to represent the Norwegian Agency for Development Cooperation (Norad), and managed by Mott Macdonald. The funders had no role in study design, data collection and analysis, decision to publish, or preparation of the manuscript.

**Competing interests:** The authors have declared that no competing interests exist.

(adjusted PR = 1.03, 95% CI 1.00–1.06 for each year), neighbourhood, and having bought sex from men (adjusted PR = 1.53, 95% CI 1.12–2.08).

## Conclusions

HIV prevalence was very high. Viral suppression among those HIV-positive was similar to the general male population in South Africa, but remains far short of national and international targets. A majority of HIV-negative MSM/TG had HIV tested in the previous 6 months, though there is room for improvement.

## Introduction

HIV surveillance and programming targeting gay and bisexual men, other men who have sex with men and transgender women (MSM/TG) have until recently been neglected in countries with generalised HIV epidemics[1]. Over the last ten years, this has begun to change, with increasing funder and policymaker attention paid to the burden of HIV experienced by these populations[2–4], including in South Africa's National Strategic Plan for HIV, TB and STI's [5]. Over this same period, coverage of antiretroviral treatment (ART) in sub-Saharan Africa has greatly expanded and AIDS-related mortality in general populations has fallen dramatically[6, 7]. Effective treatment to achieve viral suppression also avoids sexual transmission of the virus, including among MSM[8], and at the population level, 'Treatment as Prevention' (TasP) has the potential to reduce HIV incidence[9, 10]. Accordingly, UNAIDS set 'care cascade' targets specifying that of all HIV-positive individuals in a given population, 90% should be aware of their status, 90% of those aware should be on antiretroviral treatment (ART) and 90% of those on ART should be virally suppressed by 2020[11].

Targets have not yet been reached in South Africa: among all males aged 15 years and above in Gauteng Province, 85% of those HIV-positive were estimated to be aware of their status, 43% on ART and 34% virally suppressed in 2017[12]. A modelling study for Johannesburg estimated that among people aged 15 and older living with HIV in 2016, 73% were aware of their status, 45% on ART and 26% virally suppressed[13], indicating large gaps between diagnosis and access to treatment. It is possible, however, that the cascade for MSM and transgender populations differs substantially, but to date little empirical population-representative evidence from sub-Saharan Africa has been published to offer insight[14–18].

Experiences of violence, harassment, and stigma routinely inhibit MSM/TG's access to healthcare[19, 20]. While same-gender sex is legally protected in South Africa, MSM/TG nevertheless face societal discrimination[21]. There is concern that stigma and discrimination toward MSM/TG may pose barriers to prompt diagnosis, linkage to and retention in care, and adherence to ART, leading to worse HIV care outcomes. Stigma might act differentially across the cascade; there could be more pressure to disclose sexual orientation at the point of HIV testing than at the point of treatment initiation or follow-up. A study of MSM and other men attending two 'Health4Men' MSM-friendly clinics in Johannesburg found no evidence for a difference in ART retention after 24 months among men reporting same sex behaviour and those not[22], but it is unknown whether the same would be found for clinics not specifically tailoring for the needs of MSM/TG.

Prompt initiation of HIV care necessarily depends upon uptake and frequency of HIV testing, which itself may be inhibited due to stigma. MSM/TG who initially test HIV-negative need to test repeatedly during periods at risk. Men in South Africa test less frequently than

women[23], while MSM could be further inhibited due to aforementioned healthcare-related stigma. A qualitative study among urban MSM found that HIV testing was often viewed as a one-off requirement[24]. While there is historical evidence of low testing rates among South African MSM[25], testing appears to have increased between 2008 and 2013, concordant with increasing provision of MSM programming[26] and increases in HIV testing amongst men as a whole during this period[27]. There is a need now for up-to-date estimates and preferences for HIV testing among MSM/TG.

To inform HIV prevention and care programming, the TRANSFORM study set out to provide population-representative empirical estimates and correlates of the HIV care cascade and viral suppression among HIV-positive MSM/TG, and recency of and preferences for HIV testing among HIV-negative MSM/TG in Johannesburg, and to explore characteristics associated with these outcomes and behaviours.

## Methods

### Population

Eligibility criteria included being aged 18 years or older; male gender assigned at birth or current male identity; sex with a man in the past 12 months; and resident in Johannesburg.

### Sampling and recruitment

To obtain representative estimates, we recruited participants using respondent-driven sampling (RDS) [28], informed by formative research conducted within the TRANSFORM study. Recruitment proceeded in 'waves' with two coupons valid for two weeks issued to each participant, except when nearing the target sample size when we reduced to one coupon. Participants were reimbursed ZAR 150 for participation in the study and received ZAR 150 for each recruitee.

### Data collection

Recruitment began in April 2017 with four seed participants, with five additional seeds added until September 2017. Following eligibility screening and informed consent, participants privately completed a self-administered survey in English or Zulu using a tablet computer (see S3 Survey instrument of S3 Data), had HIV pre-test counselling, physical examination, blood sample collection, gave swabs for STI testing with a research nurse, and received HIV/STI post-test counselling and STI treatment. Participants were given direct referrals for HIV related care or Pre Exposure Prophylaxis (PrEP) access depending on HIV test outcome, and referrals for syndromic STI treatment onsite or at a local government clinic for partners.

**HIV testing and care cascade indicators.** Participants gave a blood sample for HIV rapid-testing, ELISA-confirmed if positive (Alere Combo and Advanced Quality HIV tests, back-up Alere Determine), CD4 count and viral load-testing (GeneXpert) if HIV-positive. We considered individuals as virally suppressed if they had <50 viral copies/ml plasma, in line with current South African treatment guidelines[29]. There is also evidence that individuals with viral loads greater than 50 copies/ml plasma but less than 1000 copies/ml show increased likelihood of poor treatment outcomes compared to those with viral load measures less than 50 copies/ml[30]. To aid comparison with past studies and those from other settings, we also show the proportion virally suppressed according to cut-offs of <200 copies/ml, <400 copies/ml and <1000 copies/ml in S2 Viral suppression according to different viral load cut-offs of S2 Data.

Participants were considered aware of their HIV-positive status if they reported in the self-administered survey ever having tested for HIV and receiving a positive result at their most recent HIV test. Participants reported month and year of most recent HIV test. Assuming tests were conducted on the first of each month, together with interview date we created binary indicators for having tested in the previous three, six and twelve months. Individuals reporting that they had previously tested HIV-positive were asked if they had received care, whether they had "ever started taking ART" and whether they were "currently taking ART".

**Other indicators.** Participants reported their age in years, previous month's income, highest educational attainment, employment status, religion, population group (using South African categories), marital status, spouse's gender, neighbourhood of residence and place of birth, their sexual identity, their sex assigned at birth and their current gender identity. We considered participants 'cisgender male' if both sex assigned at birth and current gender were male; 'transfeminine' if sex assigned at birth was male and current identity was female or transgender, 'transmasculine' if sex assigned at birth was female and current identity was male/transgender and 'non-binary' if this was their current gender identity. We used the AUDIT scale[31] to assess alcohol use and asked about frequency of use of a variety of recreational drugs, reporting here on use at least once in the past month, excepting tobacco. We used the PHQ-9 scale to assess depressive symptomatology, with cut-off points at 0, 5, 10, 15 and 20 for none, mild, moderate and moderately severe and severe depression, and asked participants whether they openly talked, tried 'very hard' or 'somewhat hard' to hide that that they had sex with men with their friends, family and healthcare workers, or neither spoke openly nor tried to hide it. Participants reported aspects of their sexual behaviour in the previous 3 months separately for male and female partners (if applicable), including number of partners, sexual roles (insertive/receptive anal sex or no anal sex, oral sex with men, and vaginal or anal sex with women), frequency of condom use by role ('always', 'most of the time', 'some of the time', 'rarely', 'never', made binary always/not always), and whether they had given, or had been given money, gifts or favours for sex in the previous 12 months. Participants self-reported experiencing STI symptoms at the time of survey and in the previous 12 months (urethral, genito-urinary or rectal).

For AUDIT and PHQ-9, if participants were missing just one or two responses to the scale of questions, we replaced those missing with the mean of the individual's other responses (mean of questions 1–9 for PHQ9[32] and questions 2–8 or 9–10 for AUDIT).

**RDS recruitment diagnostics.** As recommended, recruitment was monitored biweekly and assessed against RDS assumptions[33]. We describe RDS recruitment in detail in S1 Respondent Driven Sampling recruitment diagnostics of S1 Data. Convergence of HIV status, viral suppression and HIV testing in the last 6 months were judged as reasonable when the cumulative RDS-weighted estimate stabilised prior to the full recruitment amongst HIV-positive and negative participants respectively, examined visually.

## Statistical analysis

RDS-II weighting[34] was used to account for sampling design, dropping seed participants (who were purposively selected), and weighting by inverse MSM network size, assuming this was proportional to the probability that they would be given a coupon to participate in the study. Confidence intervals were obtained via bootstrapping.

We described the sociodemographic characteristics of MSM/TG by HIV status, as tested in the study. We examined reported HIV testing behaviour among participants testing HIV-negative in the study, and the care cascade amongst those HIV-positive. We described the proportion of HIV-negative MSM/TG who had ever tested, tested in the last three, six and twelve

months, the location of and satisfaction with most recent test, and future testing preferences. We examined the associations between these preferences and age, sexual identity and gender identity characteristics using chi-squared tests with a correction for survey data[35]. As the outcome was prevalent, we used RDS-II weighted robust Poisson regression (also dropping seed participants) to obtain weighted prevalence ratios[36, 37] to assess associations between testing in the last 6 months and sociodemographic, sexual behaviour and self-reported STI symptoms, examining first bivariate associations and then adjusting sexual behaviour and STI symptom models for any sociodemographic variables showing evidence for associations with the outcome (Wald test p<0.2).

The HIV care cascade was constructed by using the RDS-II weighted percentage of all MSM/TG testing HIV-positive in the study who knew their status, were on ART and whose viral load was suppressed, and assessing these against the 90-90-90 targets[11]. We described the proportion of HIV-positive MSM/TG who were 'satisfied'/ 'very satisfied' with the privacy and respect they were accorded in their most recent HIV care interaction.

Associations with viral suppression among HIV-positive MSM/TG were also examined using RDS-II weighted robust Poisson regression, as above, first examining sociodemographic characteristics, recreational drug use, depression, disclosure and sexual behaviours adjusted only for age, and then retaining in the adjusted model those characteristics for which the Wald test p value was <0.2.

Analyses were conducted using the RDS: Respondent-Driven Sampling package version 0.9–2[38] and the survey: analysis of complex survey samples package version 3.33–2[39] for R version 3.5.1[40] and Stata version 12[41].

### Ethics statement

This study was approved by the ethics committees at the University of the Witwatersrand, and the London School of Hygiene and Tropical Medicine.

## Results

### Sociodemographic characteristics of HIV-positive and HIV-negative MSM/TG

118/300 (37.5%) MSM/TG were HIV-positive and 182/300 were HIV-negative. One seed participant refused HIV testing. Mean age was 26 years, though HIV-positive MSM/TG were older on average than those HIV-negative, Table 1. The majority were born in Johannesburg, had completed secondary education, were Black African, unemployed, Christian and unmarried. While most MSM/TG identified as cisgender male, a substantial minority were transfeminine (15.7% among those HIV-positive and 11.7% among those HIV-negative). HIV-negative MSM/TG more often identified as bisexual, heterosexual or other compared to HIV-positive MSM/TG (40.5% compared to 11.9%).

### HIV testing behaviours and preferences amongst HIV-negative MSM/TG

24/171 HIV-negative participants reported previous HIV testing but had a missing or partially completed month and year of last HIV test. We did not find statistical evidence for systematic differences between those with and without last HIV test dates (chi-squared tests p>0.1 for all characteristics shown in Table 2 and social network size, with the exception of paying a man in money, gifts or favours in exchange for sex in the last 12 months ('buying sex')). Participants reporting buying sex were more likely to have missing dates for last HIV test, 6/17 compared to 18/153, p = 0.008.

**Table 1. Sociodemographic characteristics of MSM/TG in Johannesburg by HIV status.**

| Sociodemographic Characteristics | | HIV-positive, n = 118 | | HIV-negative, n = 182 | |
|---|---|---|---|---|---|
| | | n | RDS% | n | RDS% |
| Age in years | 18–21 | 15 | 12.8 | 59 | 30.6 |
| | 20–24 | 16 | 13.4 | 51 | 30.0 |
| | 25–29 | 36 | 27.6 | 36 | 19.7 |
| | 30+ | 51 | 46.2 | 36 | 19.4 |
| Place of birth | Born in Johannesburg | 76 | 66.9 | 105 | 59.0 |
| | Born in South Africa, but not Johannesburg | 33 | 26.6 | 65 | 34.6 |
| | Born outside of South Africa | 7 | 6.6 | 11 | 6.4 |
| | Missing | 2 | | 1 | |
| Neighbourhood residence | Soweto | 66 | 58.3 | 92 | 53.6 |
| | Braamfontein | 9 | 7.0 | 18 | 7.7 |
| | Hillbrow | 16 | 15.6 | 28 | 17.1 |
| | Orange Farm | 7 | 5.4 | 6 | 3.1 |
| | Other | 20 | 13.7 | 38 | 18.6 |
| Religion | Christianity | 104 | 83.8 | 155 | 83.7 |
| | Islam | 3 | 2.4 | 1 | 1.0 |
| | None | 9 | 13.8 | 26 | 15.3 |
| | Other | 1 | 0.0 | 0 | 0.0 |
| | Missing | 0 | | 1 | |
| Education completed | Higher Education | 34 | 31.2 | 41 | 18.2 |
| | High School | 70 | 57.4 | 129 | 75.6 |
| | Junior High | 12 | 9.7 | 10 | 5.7 |
| | Primary | 1 | 1.7 | 1 | 0.3 |
| | None | 0 | 0.0 | 1 | 0.3 |
| | Missing | 1 | | 0 | |
| Employment status | Employed full-time | 16 | 13.3 | 16 | 6.1 |
| | Employed part-time | 13 | 12.5 | 29 | 16.5 |
| | Self employed | 14 | 9.2 | 19 | 10.0 |
| | Student | 3 | 2.3 | 16 | 7.1 |
| | Unemployed | 69 | 60.1 | 98 | 5.9 |
| | Other | 2 | 2.6 | 4 | 1.4 |
| | Missing | 0 | | 1 | |
| Income per month / ZAR | Under 500 | 33 | 28.6 | 49 | 27.8 |
| | 500–999 | 10 | 11.1 | 29 | 21.6 |
| | 1000–1999 | 22 | 14.5 | 35 | 22.5 |
| | 2000–4999 | 32 | 27.3 | 41 | 22.6 |
| | 5000+ | 16 | 18.6 | 14 | 5.5 |
| | Missing | 5 | | 14 | |
| Population group | Black African | 113 | 96.4 | 173 | 94.1 |
| | Coloured | 4 | 1.9 | 5 | 3.4 |
| | White | 0 | 0.0 | 2 | 0.5 |
| | Prefer not to say | 0 | 0.0 | 1 | 1.0 |
| | Other | 1 | 1.7 | 1 | 1.0 |
| Sexual Identity | Gay or homosexual | 103 | 88.1 | 112 | 59.6 |
| | Bisexual | 12 | 11.7 | 59 | 35.3 |
| | Heterosexual | 0 | 0.0 | 3 | 1.3 |
| | Other/Don't know | 1 | 0.2 | 8 | 3.9 |

(*Continued*)

**Table 1.** (Continued)

| Sociodemographic Characteristics | | HIV-positive, n = 118 | | HIV-negative, n = 182 | |
|---|---|---|---|---|---|
| | | n | RDS% | n | RDS% |
| | Missing | 2 | | 0 | |
| Gender identity | Cisgender male | 95 | 80.6 | 138 | 76.9 |
| | Transfeminine | 20 | 15.7 | 24 | 11.7 |
| | Transmasculine | 1 | 0.3 | 1 | 0.3 |
| | Non-binary | 2 | 3.4 | 19 | 11.2 |
| Marital Status | Not married | 103 | 88.2 | 163 | 88.4 |
| | Married to a man or transgender individual | 13 | 11.8 | 17 | 10.6 |
| | Married to a woman | 0 | 0.0 | 2 | 1.0 |
| | Missing | 2 | | 0 | |

n = 1 individual did not consent to HIV testing.

Few HIV-negative MSM/TG reported never previously HIV testing (10/181, 5.5%), while 100/157, 61.5%, reported having tested in the previous 6 months and 118/157, 73.0% tested in the last 12 months, Table 3.

The most common location for last HIV test was a public clinic or hospital (69/166, 47.1%), followed by a community HIV testing centre for the general public (41/166, 24.8%), while 25/166 (13.2%) had most recently tested at an MSM-specific service. Across all HIV testing locations, MSM/TG reported high levels of satisfaction with the privacy and respect they were shown. The distribution of preferences for future testing venues was diverse and similar to that of last testing location, except for a higher preference for an MSM-specific community testing facility and a private clinic or doctor compared with the percentage who had actually tested in one (19.8% compared to 13.2%, and 18.3% compared to 9.7% respectively). Just under half of participants preferred to test in a public clinic or hospital (46.9%) and preferred to have the test conducted by a doctor/clinician (42.0%), while very few expressed a preference for self-testing (6.0%).

There was little statistical evidence that testing preferences varied by sociodemographic characteristics (full results not shown), with the exception of gender identity (p = 0.006). More transfeminine individuals preferred to test in a public clinic/hospital (18/29, 69.3%) compared to cisgender (62/142, 50.3%) and non-binary (5/18, 22.9%) MSM/TG, while fewer preferred to test at a private clinic (2/29, 10.2%) compared to cisgender (28/142, 19.0%) or non-binary individuals (4/18, 30.5%). There was little difference in the percentages preferring to test in clinics targeted towards MSM by gender identity, though there was no response option given for transgender-tailored clinics.

## Associations with having HIV tested in the previous 6 months among HIV-negative MSM/TG

There was little evidence for associations between sociodemographic characteristics and having HIV tested in the previous 6 months among HIV-negative MSM/TG. Compared to HIV-negative cisgender males, transfeminine individuals were 0.57 times as likely to have tested for HIV in the previous 6 months (95% CI 0.30–1.09), but the statistical evidence for an association between testing and gender identity was not significant overall (p = 0.222).

There was strong evidence that MSM/TG who had exchanged sex for money, gifts or favours in the previous 12 months, 'sold sex', were more likely to have HIV tested in the last 6 months (PR = 1.67, 95% CI 1.36–2.05). There was weak statistical evidence that MSM/TG who

**Table 2. Associations with testing for HIV in the previous 6 months among MSM/TG testing HIV-negative in the study.**

| Characteristics of the HIV-negative MSM/TG population | | n | RDS% | Crude PR | 95% CI | | p value |
|---|---|---|---|---|---|---|---|
| **Sociodemographic Characteristics** | | | | | | | |
| Age in years | *mean not tested / tested* | 24.5 | 24.3 | 1.00 | 0.97 | 1.02 | 0.795 |
| Born in Johannesburg | Born in Johannesburg | 55/93 | 58.9 | 1.00 | | | 0.573 |
| | Born in SA, not Johannesburg | 37/53 | 69.1 | 1.17 | 0.87 | 1.58 | |
| | Born outside of SA | 8/10 | 61.6 | 1.05 | 0.54 | 2.01 | |
| Religion | None | 12/24 | 61.0 | 1.00 | | | 0.965 |
| | Christianity/Islam/Other | 88/133 | 61.6 | 1.01 | 0.68 | 1.50 | |
| Education completed | Junior High School | 6/10 | 65.0 | 1.06 | 0.61 | 1.87 | 0.973 |
| | High School | 70/111 | 61.0 | 1.00 | | | |
| | College/University/Higher Education | 24/36 | 61.3 | 1.02 | 0.71 | 1.46 | |
| Employment status | Unemployed | 59/88 | 65.3 | 1.00 | | | 0.325 |
| | Employed or student | 41/69 | 55.8 | 0.85 | 0.63 | 1.17 | |
| Income in the last month | 0–499 | 26/41 | 55.8 | 1.00 | | | 0.596 |
| (ZAR) | 500–999 | 13/21 | 68.8 | 1.23 | 0.78 | 1.94 | |
| | 1000–1999 | 19/31 | 51.4 | 0.92 | 0.55 | 1.56 | |
| | 2000–4999 | 27/37 | 65.9 | 1.18 | 0.77 | 1.82 | |
| | 5000+ | 4/14 | 41.3 | 0.74 | 0.32 | 1.71 | |
| Sexual Identity | Bisexual, heterosexual, other | 36/65 | 60.8 | 1.00 | | | 0.846 |
| | Gay or homosexual | 64/92 | 62.0 | 0.97 | 0.72 | 1.31 | |
| Gender identity | Cisgender male | 81/120 | 64.4 | 1.00 | | | 0.222 |
| | Transfeminine | 9/19 | 36.9 | 0.57 | 0.30 | 1.09 | |
| | Transmasculine | 0/1 | 0.0 | - | - | - | |
| | Non-binary | 10/17 | 67.1 | 1.04 | 0.68 | 1.59 | |
| Marital Status | Single/divorced/widowed | 90/141 | 60.0 | 1.00 | | | 0.274 |
| | Married to a man or transgender individual | 8/12 | 74.0 | 1.23 | 0.85 | 1.80 | |
| | Married to a woman | 2/2 | 100.0 | - | - | - | |
| Neighbourhood | Soweto | 51/83 | 63.1 | 1.00 | | | 0.524 |
| | Hillbrow | 15/19 | 74.6 | 1.18 | 0.81 | 1.72 | |
| | Braamfontein | 14/18 | 66.9 | 1.06 | 0.65 | 1.73 | |
| | Orange Farm | 4/5 | 65.1 | 1.03 | 0.45 | 2.35 | |
| | Other* | 16/32 | 44.7 | 0.71 | 0.43 | 1.17 | |
| **Sexual behaviours** | | | | | | | |
| Sex with a man (3months) | No | 19/41 | 47.0 | 1.00 | | | 0.083 |
| | Yes | 81/116 | 67.6 | 1.44 | 0.96 | 2.16 | |
| Sex with a woman (3 months) | No | 61/94 | 56.5 | 1.00 | | | 0.239 |
| | Yes | 39/63 | 67.3 | 1.19 | 0.89 | 1.59 | |
| 2+ male partners (3 months) | No | 83/131 | 63.0 | 1.00 | | | 0.433 |
| | Yes | 17/26 | 51.4 | 0.82 | 0.49 | 1.35 | |
| Sell sex to men (12 months) | No | 78/129 | 55.1 | 1.00 | | | <0.001 |
| | Yes | 22/27 | 92.0 | 1.67 | 1.36 | 2.05 | |
| Buy sex from a man (12 months) | No | 91/144 | 61.1 | 1.00 | | | 0.313 |
| | Yes | 8/11 | 76.1 | 1.25 | 0.81 | 1.91 | |
| Condomless Anal Intercourse with a man (3 months) | No | 61/104 | 58.8 | 1.00 | | | 0.383 |
| | Yes | 39/53 | 67.1 | 1.14 | 0.85 | 1.54 | |

(*Continued*)

**Table 2.** (Continued)

| Characteristics of the HIV-negative MSM/TG population | | n | RDS% | Crude PR | 95% CI | | p value |
|---|---|---|---|---|---|---|---|
| Sexual role with men during anal sex, (3 months) | No anal sex with men | 23/45 | 53.0 | 0.68 | 0.45 | 1.03 | 0.262 |
| | Receptive anal intercourse | 27/39 | 56.4 | 0.73 | 0.47 | 1.13 | |
| | Insertive anal intercourse | 35/53 | 67.2 | 0.86 | 0.61 | 1.22 | |
| | Versatile | 15/20 | 77.8 | 1.00 | | | |
| Self-reported STI symptoms (12 months) | No | 82/123 | 66.2 | 1.00 | | | 0.094 |
| | Yes | 18/34 | 44.5 | 0.67 | 0.42 | 1.07 | |

n = 157 HIV-negative with responses to ever tested and date of last test.

n's do not add to total where there are some missing responses.

All models exclude seed participants and include weighting for inverse network size (RDS-II).

There was not evidence for associations (Wald test p<0.2) between sociodemographic characteristics and testing, so we show only univariate associations.

*Other neighbourhoods include those within Johannesburg with fewer than 10 participants in the study sample.

had had sex with a man in the previous 3 months were more likely to have tested in the last 6 months compared to those who had not (PR = 1.44, 95% CI 0.96–2.16), and that those who reported experiencing STI symptoms in the past year were less likely to have tested for HIV in the previous 6 months, (PR = 0.67, 95% CI 0.42–1.07).

## Viral suppression and engagement in treatment and care among HIV-positive MSM/TG

The percentage of HIV-positive MSM/TG who were virally suppressed (<50 copies/ml) was estimated to be 46.9% (95% CI 31.5–62.3%, n = 58/118), Fig 1, Table 4. Self-reported knowledge of HIV-positive status was estimated to be 56.5% (95% CI 40.4–72.6%, n = 76/118) and current ART coverage 30.0% (95% CI 17.1–43.0%, n = 39/118) of all those HIV-positive. Of those who reported currently taking ART, there were 10/39 who were not virally suppressed. There were 29 participants who tested positive for HIV and were virally suppressed but who either did not self-report ever testing positive for HIV, or did not self-report current ART use.

Among HIV-positive MSM/TG, 75.6% had last attended HIV care at public clinics and 18.1% at MSM-specific clinics, with the remaining at private clinics. There were 89.3% (n = 63/90) who were satisfied with their last clinic's privacy and 91.5% (n = 65/70) who were satisfied with the respect they were shown; these figures were 100% among those visiting MSM-specific clinics.

## Factors associated with viral suppression amongst HIV-positive MSM/TG

Being virally suppressed was associated in crude analyses with increasing age, PR = 1.03 for each year, 95% CI 0.99–1.06, Table 5. HIV-positive MSM/TG living in Braamfontein, a central neighbourhood, as compared to Soweto, were less likely to be virally suppressed, though the numbers were small (1/9 compared to 39/66, PR = 0.04, 95% CI 0.00–0.33). MSM/TG who were not open with their family about the fact that they had sex with men were less likely to be virally suppressed, though the evidence for the association was weak (PR = 0.72, 95% CI 0.46–1.14). There was evidence that MSM/TG reporting recreational drug use in the last month were more likely to be virally suppressed (PR = 1.56, 95% CI 1.00–2.44). Having paid a man in money, gifts or favours in exchange for sex in the past 12 months was strongly associated with being virally suppressed compared to not buying sex in the last 12 months (PR = 2.16, 95% 1.61–2.89).

**Table 3. HIV testing history and experiences amongst MSM/TG testing HIV-negative in the study.**

| Among all testing HIV-negative, n = 182 | | | | | | |
|---|---|---|---|---|---|---|
| **Testing history** | **n/N** | **RDS %** | | | | |
| Ever had an HIV test | 171/181 | 94.5 | | | | |
| Had an HIV test in the previous 12 months | 118/157* | 73.0 | | | | |
| Had an HIV test in the past 6 months | 100/157* | 61.5 | | | | |
| Had an HIV test in the past 3 months | 77/157* | 49.8 | | | | |
| | | | **'Satisfied' or 'Very satisfied' with privacy** | | **'Satisfied' or 'Very satisfied' with respect shown** | |
| **Testing preferences** | **n/N** | **RDS %** | **n** | **RDS %** | **n** | **RDS %** |
| **Where last tested for HIV, of those ever testing**** | | | | | | |
| Public hospital or clinic | 69/167 | 47.1 | 68 | 97.7 | 66 | 98.7 |
| Private hospital or clinic | 20/167 | 9.7 | 19 | 98.1 | 19 | 98.1 |
| Community HIV testing service for the public | 41/167 | 24.9 | 37 | 92.7 | 40 | 99.1 |
| Community HIV testing service for MSM only | 25/167 | 13.2 | 24 | 95.8 | 24 | 95.8 |
| At home | 3/167 | 2.4 | 3 | 100 | 3 | 100 |
| University/school (write-in response) | 7/167 | 2.4 | 7 | 100 | 7 | 100 |
| Research study (write-in response) | 2/167 | 0.4 | 2 | 100 | 2 | 100 |
| *Amongst those HIV negative and reporting that their last test was negative, n = 163****: | | | | | | |
| **Where would you choose to test in future?** | | | | | | |
| Public hospital or clinic | 69/162 | 46.9 | | | | |
| Private hospital or clinic | 27/162 | 18.3 | | | | |
| Community HIV testing service for the public | 17/162 | 8.8 | | | | |
| Community HIV testing service for MSM only | 39/162 | 19.8 | | | | |
| At home | 10/162 | 6.2 | | | | |
| **Who would you prefer to perform an HIV test in future?** | | | | | | |
| Doctor or clinical officer | 62/162 | 42.1 | | | | |
| Nurse | 27/162 | 17.0 | | | | |
| Counsellor | 19/162 | 12.4 | | | | |
| MSM community worker | 44/162 | 22.6 | | | | |
| Me (i.e., self-test) | 10/162 | 6.0 | | | | |

*n = 24 responses to last HIV test date were missing, n = 1 missing ever tested response.

**n = 5 responses to where last tested were missing.

***n = 1 response to where would like to test in future missing.

In adjusted analyses, the evidence that older MSM/TG were more likely to be virally suppressed remained (aPR = 1.03, 96% CI 1.00–1.06), while the statistical evidence for neighbourhood associations weakened somewhat (p = 0.070). The association between viral suppression and recreational drug use also weakened (aPR = 1.36, 95% CI 0.89–2.06, p = 0.155). Strong evidence that MSM who had bought sex were more likely to virally suppressed remained, though the strength of the association reduced somewhat (aPR = 1.53, 95% CI 1.12–2.08).

## Discussion

We estimated a very high HIV prevalence among MSM/TG in Johannesburg, and found half of those HIV-positive to be virally unsuppressed. These findings indicate gaps in the HIV care cascade that represent missed opportunities to improve the health of HIV-positive MSM/TG and to prevent ongoing HIV transmission. Younger MSM/TG who were HIV-positive were

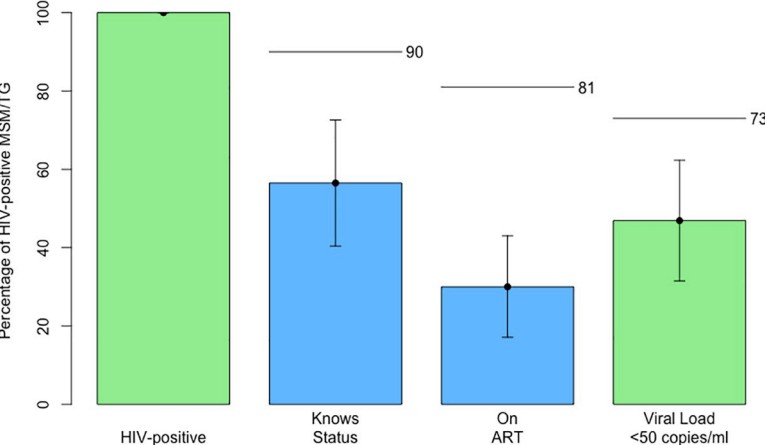

**Fig 1. The HIV care cascade among MSM/TG in Johannesburg, n = 300.**

less likely to be virally suppressed, emphasising the need for regular and accessible testing and support to engage in HIV care for this group.

We found a higher proportion of MSM/TG to be virally suppressed at 47% (32–62%) than the 34% estimated among all adult males living with HIV in Gauteng in 2017 [12], the 26% estimated in Johannesburg in 2016 [13] and similar to the 47% (95% CI 43–52%) estimated to be virally suppressed among the HIV-positive adult population in South Africa as a whole in 2017[42]. This was the case even with our lower viral load cut-off of 50 copies/ml plasma as compared to 400/ml plasma. Our estimate is also higher than modelled estimates for virological suppression among South African MSM (22%, data from 2013–2015)[43]. It is plausible that in addition to challenges related to having sex with men, HIV-positive MSM/TG experience some of the same barriers to achieving viral suppression that men as whole do, as estimates for viral suppression among women are higher than those of men overall (43% compared to 58% nationally among adults[42]). Knowledge of status and current ART use among HIV-positive MSM/TG were much higher than that reported amongst MSM in two districts of neighbouring Mpumalanga province in 2012–2013, (56.5% compared to 28.1% and

**Table 4. The HIV care cascade among MSM/TG in Johannesburg, n = 300.**

|  |  | n | Unweighted sample % | RDS-II weighted % | 95% CI |  |
|---|---|---|---|---|---|---|
| **Of All HIV-Positive, n = 118** |  |  |  |  |  |  |
| Knows HIV-Positive Status | no | 42 |  |  |  |  |
|  | yes | 76 | 64.4 | 56.5 | 40.4, | 72.6 |
| Currently on ART of all HIV-Positive | no | 79 |  |  |  |  |
|  | yes | 39 | 33.1 | 30.0 | 17.1, | 43.0 |
| Virally suppressed of all HIV-positive | no: 50+ copies/ml | 60 |  |  |  |  |
|  | yes: <50 copies/ml | 58 | 49.2 | 46.9 | 31.5, | 62.3 |
| **Expressed as 90-90-90** |  |  |  |  |  |  |
| Knows HIV-Positive Status | no | 42 |  |  |  |  |
|  | yes | 76 | 64.4 | 56.5 | 40.4, | 72.6 |
| Currently on ART of those who know they are positive | no | 37 |  |  |  |  |
|  | yes | 39 | 51.3 | 53.2 | 32.7, | 73.7 |
| Virally suppressed of those on ART | no: 50+ copies/ml | 10 |  |  |  |  |
|  | yes: <50 copies/ml | 29 | 74.4 | 77.3 | 51.4, | 100.0 |

**Table 5. Associations with viral suppression among HIV-positive MSM/TG.**

| Characteristics of the HIV-positive MSM/TG population | | n | RDS% | Crude PR | 95% CI | | p value | *aPR | 95% CI | | p value |
|---|---|---|---|---|---|---|---|---|---|---|---|
| **Sociodemographic characteristics, sexual and gender identity** | | | | | | | | | | | |
| Age in years | *Mean unsuppressed / suppressed* | 28.1 | 30.2 | 1.03 | 0.99 | 1.06 | 0.130 | 1.03 | 1.00 | 1.06 | 0.051 |
| Income per month / ZAR | Under 500 | 17/33 | 42.5 | 1.00 | | | 0.869 | | | | |
| | 500–999 | 6/10 | 48.9 | 1.15 | 0.48 | 2.73 | | | | | |
| | 1000–1999 | 10/22 | 41.0 | 0.96 | 0.43 | 2.14 | | | | | |
| | 2000–4999 | 17/32 | 56.8 | 1.34 | 0.72 | 2.47 | | | | | |
| | 5000+ | 7/16 | 51.2 | 1.20 | 0.58 | 2.51 | | | | | |
| Born in Johannesburg | Born in Johannesburg | 43/76 | 47.6 | 1.00 | | | 0.631 | | | | |
| | Born in SA, not Johannesburg | 12/33 | 43.8 | 0.92 | 0.53 | 1.61 | | | | | |
| | Born outside of SA | 3/7 | 65.2 | 1.37 | 0.66 | 2.87 | | | | | |
| Religion | None | 3/10 | 24.3 | 1.00 | | | 0.226 | | | | |
| | Christianity/Islam/Other | 55/108 | 50.7 | 2.09 | 0.64 | 6.81 | | | | | |
| Education completed | Up to Junior High School | 8/13 | 50.0 | 1.07 | 0.52 | 2.20 | 0.985 | | | | |
| | High School | 32/70 | 46.9 | 1.00 | | | | | | | |
| | College/University/HE | 18/34 | 47.0 | 1.00 | 0.59 | 1.71 | | | | | |
| Employment status | Unemployed | 33/69 | 46.0 | 1.00 | | | 0.768 | | | | |
| | Employed full-time, part-time, student | 25/48 | 49.4 | 1.07 | 0.67 | 1.72 | | | | | |
| Sexual Identity | Gay or homosexual | 50/103 | 52.0 | 1.00 | | | 0.738 | | | | |
| | Bisexual, heterosexual or other | 7/13 | 46.3 | 1.12 | 0.57 | 2.20 | | | | | |
| Gender Identity | Cisgender male | 47/95 | 48.8 | 1.00 | | | 0.477 | | | | |
| | Transfeminine | 10/20 | 46.1 | 0.94 | 0.52 | 1.72 | | | | | |
| | Transmasculine | 1/1 | 100.0 | 2.05 | 1.58 | 2.64 | | | | | |
| | Non-binary | 0/2 | 0.0 | | | | | | | | |
| Marital Status | Single/divorced/widowed | 50/103 | 45.8 | 1.00 | | | 0.515 | | | | |
| | Married/civil union/legal to a man or transgender individual | 7/13 | 56.8 | 1.24 | 0.65 | 2.35 | | | | | |
| | Married to a woman | 0/0 | 0 | - | - | - | | | | | |
| Neighbourhood | Soweto | 39/66 | 55.8 | 1.00 | | | 0.032 | 1.00 | | | 0.070 |
| | Hillbrow | 6/16 | 43.2 | 0.77 | 0.38 | 1.58 | | 0.67 | 0.32 | 1.39 | |
| | Braamfontein | 1/9 | 2.2 | 0.04 | 0.00 | 0.33 | | 0.05 | 0.01 | 0.48 | |
| | Orange Farm | 5/7 | 58.2 | 1.04 | 0.44 | 2.46 | | 0.90 | 0.47 | 1.71 | |
| | Other | 7/20 | 31.5 | 0.56 | 0.26 | 1.22 | | 0.62 | 0.29 | 1.32 | |
| **Mental health, alcohol and substance use, disclosure of sexuality** | | | | | | | | | | | |
| PHQ9 | None | 29/57 | 47.4 | 1.00 | | | 0.924 | | | | |
| | Mild depression | 10/29 | 39.1 | 0.82 | 0.42 | 1.61 | | | | | |
| | Moderate depression | 11/18 | 56.7 | 1.20 | 0.65 | 2.20 | | | | | |
| | Moderately severe or severe depression | 4/8 | 56.4 | 1.02 | 0.38 | 2.75 | | | | | |
| | Missing >2 responses to scale questions | 4/6 | 45.1 | 0.95 | 0.33 | 2.71 | | | | | |
| AUDIT | Low risk drinking | 28/53 | 50.2 | 1.00 | | | 0.632 | | | | |
| | In excess of low risk drinking guidelines | 19/41 | 44.7 | 0.89 | 0.52 | 1.54 | | | | | |
| | Harmful or hazardous drinking | 4/10 | 36.0 | 0.72 | 0.24 | 2.11 | | | | | |
| | Alcohol dependence | 3/8 | 24.3 | 0.48 | 0.15 | 1.60 | | | | | |
| | Missing >2 responses to scale questions | 4/6 | 64.3 | 1.28 | 0.62 | 2.65 | | | | | |

(*Continued*)

**Table 5.** (Continued)

| Characteristics of the HIV-positive MSM/TG population | | n | RDS% | Crude PR | 95% CI | | p value | *aPR | 95% CI | | p value |
|---|---|---|---|---|---|---|---|---|---|---|---|
| Use of at least one substance in last month | No | 37/81 | 39.9 | 1.00 | | | 0.052 | 1.00 | | | 0.155 |
| | Yes | 21/37 | 62.4 | 1.56 | 1.00 | 2.44 | | 1.36 | 0.89 | 2.06 | |
| Openly talk about having sex with men with friends | Yes | 38/77 | 50.5 | 1.00 | | | 0.429 | | | | |
| | No | 19/40 | 41.0 | 0.81 | 0.48 | 1.36 | | | | | |
| Openly talk about having sex with men with family | Yes | 26/43 | 58.5 | 1.00 | | | 0.169 | 1.00 | | | 0.203 |
| | No | 30/72 | 42.4 | 0.72 | 0.46 | 1.14 | | 0.75 | 0.49 | 1.16 | |
| Openly talk about having sex with men with healthcare workers | Yes | 39/77 | 45.2 | 1.00 | | | 0.704 | | | | |
| | No | 18/40 | 49.6 | 1.10 | 0.68 | 1.79 | | | | | |
| **Sexual Behaviours** | | | | | | | | | | | |
| Sex with a man (3months) | No | 5/22 | 23.8 | 1.00 | | | 0.119 | 1.00 | | | 0.203 |
| | Yes | 53/96 | 52.4 | 2.20 | 0.82 | 5.86 | | 1.53 | 0.61 | 3.86 | |
| Sex with a woman (3 months) | No | 53/107 | 47.1 | 1.00 | | | 0.941 | | | | |
| | Yes | 5/11 | 45.7 | 0.97 | 0.44 | 2.14 | | | | | |
| 2+ male partners (3 months) | No | 37/78 | 47.3 | 1.00 | | | 0.910 | | | | |
| | Yes | 23/40 | 45.9 | 0.97 | 0.59 | 1.61 | | | | | |
| Sell sex to men (12 months) | No | 37/78 | 46.8 | 1.00 | | | 0.860 | | | | |
| | Yes | 20/37 | 48.9 | 1.05 | 0.63 | 1.73 | | | | | |
| Buy sex from a man (12 months) | No | 47/102 | 43.0 | 1.00 | | | <0.001 | 1.00 | | | 0.008 |
| | Yes | 11/14 | 92.8 | 2.16 | 1.61 | 2.89 | | 1.53 | 1.12 | 2.08 | |
| Condomless Anal Intercourse with a man (3 months) | No | 35/68 | 51.2 | 1.00 | | | 0.350 | | | | |
| | Yes | 23/50 | 40.3 | 0.79 | 0.48 | 1.30 | | | | | |
| Sexual role with men during anal sex, (3 months) | No anal sex with men last 3 months | 5/23 | 22.8 | 0.36 | 0.13 | 1.01 | 0.239 | | | | |
| | Receptive anal intercourse | 26/46 | 45.7 | 0.72 | 0.42 | 1.24 | | | | | |
| | Insertive anal intercourse | 11/20 | 53.0 | 0.84 | 0.46 | 1.54 | | | | | |
| | Versatile | 13/23 | 63.2 | 1.00 | | | | | | | |

All models exclude seed participants and include weighting for inverse network size (RDS-II).

Adjusted models were adjusted for age and those variables found to be p value <0.2 in the crude analysis. p values are from Wald tests.

n's do not add to total where there are some missing responses.

*Other neighbourhoods include those within Johannesburg with fewer than 10 participants in the study sample.

14.1% aware of status, 30% versus 14% and 10% on ART)[44]. Our higher estimates could be related to having an urban sample of MSM/TG and to our more recent data collection. Nonetheless, the percentage of HIV-positive MSM/TG whom we found to be virally suppressed remains well below the UNAIDS target of 73% of all those HIV-positive, indicating gaps in the HIV care cascade. It is encouraging however that HIV-positive MSM/TG receiving HIV care, 76% of whom were attending public clinics and hospitals, reported high levels of satisfaction with the service they received.

We found that HIV-positive MSM/TG who had bought sex from a man in the past 12 months were more likely to be virally suppressed. Our findings could possibly reflect a higher recognition of risk and thus increased care engagement among those buying sex, but we are not able to determine causality. Studies among other MSM/TG populations have found that substance use, alcohol use and mental health are associated with worse care cascade outcomes [45], and the fact that we did not find statistically significant associations with these

characteristics should be treated with caution. Our sub-sample was small, statistical error remains a possibility and we have not disaggregated different types of drug use.

Almost two thirds of HIV-negative MSM/TG reported having HIV tested within the previous 6 months, higher than among MSM in Soweto in 2008 (28%)[24, 25] and MSM in Mpumalanga in 2014–2015 prior to a testing intervention (38%)[46]. Encouragingly, testing was associated with some indicators of higher risk, including selling sex to men, though not self-reported STI symptoms. There remains a need for sustained HIV/STI testing messages. Transfeminine individuals were less likely than cisgender male individuals to have HIV tested, similar to findings from Cape Town[47], though the statistical evidence for an association between testing and gender identity was weak; further investigation is needed. Of MSM/TG who did attend HIV testing, it is encouraging that high levels of privacy and respect at last testing experience were reported. That there was not a clear majority preference for testing location and person performing the tests underscores the need for a high volume and variety of accessible and acceptable services for a diverse MSM/TG population. While few MSM/TG reported a preference for HIV self-testing in our study, it is possible that this stems from unfamiliarity: a recent study of MSM in Mpumalanga found high levels of satisfaction with HIV self-testing and increased testing frequency amongst those who had tried it[46], though studies from elsewhere have also found that men are reluctant to receive a diagnosis on their own[48].

## Limitations

Our study set out to obtain population-representative estimate of HIV testing and the care cascade among MSM/TG in Johannesburg, rather than relying upon modelled estimates. While we found that study outcomes of HIV status, viral load and testing and many sociodemographic characteristics showed evidence of converging well, there were some challenges in achieving a representative sample of MSM/TG (S1 Data), as in other MSM surveys from the continent[49, 50]. Compared to another study of MSM in Soweto from 2008, we found a lower proportion of MSM/TG who identified as heterosexual or 'straight'[25], and our estimates of sexual identity had not converged, S1 Fig A2e of S1 Data, likely indicating a difficulty in sampling this group. A higher proportion of participants described themselves as Black African compared to the population of Johannesburg overall; recruitment chains originating from White seeds did not achieve more than two sample waves and White and Asian individuals were not recruited by others. Most participants were recruited from four of the nine seeds.

Following stratification by HIV status, we might have had limited power for investigating factors that are associated with viral suppression among HIV-positive MSM/TG or recent HIV testing among HIV-Negative MSM/TG.

A further limitation was our reliance upon self-reported measures of HIV testing/status awareness and ART use. That half of MSM/TG verified to be virally suppressed did not report currently taking ART and/or did not report being aware of their status suggests that diagnosis and current ART use were under-reported among men testing HIV-positive in our study, discrepancies that we are investigating further[51]. Other surveys and studies using self-reported care cascade indicators have also pointed out this problem[52–56], with social desirability biases and survey misunderstandings posited as possible explanations. Confusion from care interactions have also been reported in South Africa[57]. Misinterpretations about the distinction between 'undetectable' viral load and 'negative' HIV status in provider interactions may represent an increasingly importance source of misclassification that should be investigated further. While our estimate of viral suppression is biologically ascertained, these reporting biases limit our ability to guide programming in identifying whether the gaps in diagnosis versus initiation and retention on ART are most significant.

## Conclusions

It is encouraging that MSM/TG do not appear to be lagging in the HIV care cascade compared to the general male population in Johannesburg and are reporting positive care experiences. However, given their higher risks as a group and very high HIV burden, there is a need to continue to focus resources towards further improvement, particularly among younger MSM/TG. A better understanding of sexual mixing and transmission patterns could aid in targeting, development of a differentiated care approach to serve the diversity of MSM would be appropriate, and a dedicated study to understand the needs of the transgender population is warranted.

## Supporting information

**S1 Data. Respondent driven sampling recruitment diagnostics.**
(DOCX)

**S2 Data. Viral suppression according to different viral load cut-offs.**
(DOCX)

**S3 Data. Survey instrument.**
(PDF)

## Acknowledgments

We thank and acknowledge the participants of the TRANSFORM study, and are grateful to assistance from the ANOVA Health Institute, Ten81 and SOHACA.

Membership of the TRANSFORM Study Group includes: Thesla Palanee-Phillips[*1], Joshua Kimani[2,3], Peter Weatherburn[4], Adam Bourne[5], Adrian D Smith[6], Elizabeth Fearon[7], Will Nutland[4], Siyanda Tenza[1], Rhoda Kabuti[3], Kerushini Moodley[1], Cecilia Mokoena[1], Jennifer Liku[3], Anthony Tukai[3], Hellen Babu[3], Amos Mugambi[3], Chrispo Nyamweya[3], Mary Kung'u[3], Polly Ngurukiri[3], Peter Muthoga[3], Erastus Irungu[3]

1. Wits Reproductive Health and HIV Institute, Johannesburg, South Africa

2. Department of Community Health Sciences, University of Manitoba, Winnipeg, Canada

3. Partners for Health and Development in Africa, Nairobi, Kenya

4. Department of Public Health, Environments and Society, London School of Hygiene and Tropical Medicine, London, United Kingdom

5. Australian Research Centre in Sex, Health and Society, LaTrobe University, Melbourne, Australia

6. Nuffield Department of Population Health, University of Oxford, Oxford, United Kingdom

7. Department of Global Health and Development, London School of Hygiene and Tropical Medicine, London, United Kingdom

[*]Principal Investigator: tpalanee@wrhi.ac.za

## Author Contributions

**Conceptualization:** Elizabeth Fearon, Adrian D. Smith, Adam Bourne, Peter Weatherburn, Thesla Palanee-Phillips.

**Data curation:** Elizabeth Fearon, Siyanda Tenza, Cecilia Mokoena, Kerushini Moodley.

**Formal analysis:** Elizabeth Fearon.

**Investigation:** Adam Bourne, Peter Weatherburn, Thesla Palanee-Phillips.

**Methodology:** Elizabeth Fearon.

**Supervision:** Peter Weatherburn, Thesla Palanee-Phillips.

**Writing – original draft:** Elizabeth Fearon.

**Writing – review & editing:** Elizabeth Fearon, Siyanda Tenza, Cecilia Mokoena, Kerushini Moodley, Adrian D. Smith, Adam Bourne, Peter Weatherburn, Thesla Palanee-Phillips.

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
