## [Decision Letter · Decision Letter 0]

29 Nov 2019

PONE-D-19-20875

HIV testing, care and viral suppression among men who have sex with men and transgender individuals in Johannesburg, South Africa

PLOS ONE

Dear Dr Smith,

Thank you for submitting your manuscript to PLOS ONE. After careful consideration, we feel that it has merit but does not fully meet PLOS ONE’s publication criteria as it currently stands. Therefore, we invite you to submit a revised version of the manuscript that addresses the points raised during the review process.

I invite you to revise and resubmit your manuscript. Please attend to the reviewer's comments, which I believe may help to better strengthen the contributions of this manuscript to the field and public health practice. In addition, please attend to the following issues I noted in my review of the manuscript:

If citation 31 is published then please include a brief statement that reviews the RDS recruitment process (methods) and challenges reaching a representative sample using RDS (discussion). It is not enough to just point the reader to a subsequent manuscript. If this paper remains unpublished by the time of your re-submission, please provide sufficient detail on the RDS method and appropriate data on the representativeness of the sample recruited.Please indicate the variables missing data in Table 1 as done in Table 2 in the foot notePlease include the test statistics for comparison of recent HIV tests with and without missing date information in text. I t is OK that these data are not presented in any of the tables.

join the reviewers in their enthusiasm for the data you present on an important, but understudied population; highlighting both areas where some successes have been achieved and areas in need of targeted intervention efforts. Please provide a point by point response to each reviewers' response, and indicate in the response how it was addressed. This will help to facilitate a more timely review.

We would appreciate receiving your revised manuscript by Jan 13 2020 11:59PM. To enhance the reproducibility of your results, we recommend that if applicable you deposit your laboratory protocols in protocols.io, where a protocol can be assigned its own identifier (DOI) such that it can be cited independently in the future. For instructions see: http://journals.plos.org/plosone/s/submission-guidelines#loc-laboratory-protocols

We look forward to receiving your revised manuscript.

Kind regards,

Laramie Smith, PhD

Academic Editor

PLOS ONE

Journal Requirements:

1. Please include a copy of the interview guide used in the study, in both the original language and English, as Supporting Information, or include a citation if it has been published previously.

4. One of the noted authors is a group or consortium TRANSFORM Study Group. In addition to naming the author group, please list the individual authors and affiliations within this group in the acknowledgments section of your manuscript. Please also indicate clearly a lead author for this group along with a contact email address.

Reviewers' comments:

Reviewer's Responses to Questions

**Comments to the Author**

1. Is the manuscript technically sound, and do the data support the conclusions?

Reviewer #1: Yes

Reviewer #2: Yes

2. Has the statistical analysis been performed appropriately and rigorously? 

Reviewer #1: Yes

Reviewer #2: No

3. Have the authors made all data underlying the findings in their manuscript fully available?

Reviewer #1: No

Reviewer #2: No

4. Is the manuscript presented in an intelligible fashion and written in standard English?

Reviewer #1: Yes

Reviewer #2: Yes

5. Review Comments to the Author

Reviewer #1: This study uses RDS methodology to obtain a representative sample of MSM/TG in Johannesburg, in order to investigate: HIV prevalence, HIV testing behaviour in HIV negative individuals and viral suppression in HIV positive individuals. The study found a high prevalence of HIV, that the majority of negative MSM/TG had tested within the last 6 months, and poor viral suppression in PLHIV, although comparable to that of men in South Africa in general.

This is an important study adding to the evidence base attempting to understand HIV in this key population which has been historically neglected in Sub-Saharan Africa.

I believe the manuscript could be improved by addressing the following comments:

1) Why was a cut-off of 200 used for viral suppression? The cut-off in South African treatment guidelines at the time of the study, as well as National Department of Health programme monitoring, is 400. The other commonly used cut-off at this time (2017) was 1000 (PEPFAR routine reporting; 2017 South African national HIV prevalence, incidence, behaviour and communication survey). In the new guidelines, 50 will be used. It would be more comparable to the other evidence and applicable to the South African setting to use 400 as a cut-off instead.

2) In South Africa it is known that there is a large group of MSM who identify as straight (example Lane et al. High HIV Prevalence Among Men who have Sex with Men in Soweto South Africa: Results from the Soweto's Men's Study AIDS Behav (2011) 15:626–634). The RDS methodology does not appear to have penetrated into this group well (not unexpected). I think it is important to mention this in the discussion.

3) Although the viral suppression rates are similar to those of other men, the rate of viral suppression in men in South Africa is well below that of women. e.g. 2017 South African national HIV prevalence, incidence, behaviour and communication survey: viral suppression in men- 43%; viral suppression in women- 58%. Is it possible that the finding of similar viral suppression in MSM/TG is related to the fact that men in general experience such significant access barriers?

4) Reference 31: has this manuscript been accepted for publication yet? I do not believe it can be referenced as it stands. From the journal website:

"Do not cite the following sources in the reference list: Unavailable and unpublished work, including manuscripts that have been submitted but not yet accepted (e.g., “unpublished work,” “data not shown”). Instead, include those data as supplementary material or deposit the data in a publicly available database."

5) There are a few minor editing comments:

- line 103: please change "18 years plus" to "18 years or older"

- Table 1, employment status, unemployed- there appears to be an error in the % column

Reviewer #2: Review: HIV testing, care and viral suppression among men who have sex with men and transgender individuals in Johannesburg, South Africa

A very well written paper of major interest in the field. The authors estimated a very high HIV prevalence among MSM/TG in Johannesburg, and half of those HIV-positive were virally unsuppressed. These findings indicate gaps in the HIV care cascade that represent missed opportunities to improve the health of HIV-positive MSM/TG and to prevent ongoing HIV transmission.

I only have one major concern that requires the author’s revision before publication. The authors have conducted univariate and multivariate logistic regression to identify factors associated with HIV-testing in the previous 6 months amongst those HIV-negative; and to identify factors associated with viral suppression amongst those HIV-positive.

While the authors have correctly interpreted the odds ratios (OR). In this population, these binary outcomes (HIV testing, access to care and viral suppression) are common (>10%), it is often more desirable to estimate a relative risk (RR) or prevalence ratio since there is an increasing differential between the RR and odds ratio with increasing prevalence rates, and there is a tendency for some to interpret ORs as if they are RRs. The authors can estimate the RR in Stata with a log-binomial regression model or should there be convergence issues, apply the Poisson regression model with a robust error variance. A major consideration in public health research is need to communicate study results to a wider non-statistical community i.e. the medical community, general public, and policy makers. Poisson regression with robust variance and log-binomial regression provide correct estimates and are a better alternative for the analysis of this kind of study design with binary outcomes than logistic regression, as the relative risk or prevalence ratio is more interpretable and easier to communicate than the odds ratio.

Please see references below.

1. McNutt LA, Wu C, Xue X, Hafner JP. Estimating the Relative Risk in Cohort Studies and Clinical Trials of Common Outcomes. Am J Epidemiol 2003; 157(10):940-3.

2. Zou G. A Modified Poisson Regression Approach to Prospective Studies with Binary Data. Am J Epidemiol 2004; 159(7):702-6.

3. Sander Greenland , Model-based Estimation of Relative Risks and Other Epidemiologic Measures in Studies of Common Outcomes and in Case-Control Studies, American Journal of Epidemiology 2004;160:301-305

Simba Takuva, MBChB, MSc, DTM&H.

6. PLOS authors have the option to publish the peer review history of their article (what does this mean?). If published, this will include your full peer review and any attached files.

Reviewer #1: No

Reviewer #2: Yes: Simbarashe Takuva

---

## [Author Response · Author response to Decision Letter 0]

26 Mar 2020

February 28, 2020

Dear Laramie,

Please find attached our revised manuscript submission. 

We thank the editor and reviewers for taking the time to read and make suggestions to improve our manuscript. Below, we respond to editor and reviewer comments, and explain why it is not possible to make data publicly available but give guidance on how researchers can obtain the data upon request.

Yours sincerely,

Elizabeth Fearon

Editor Comments

I invite you to revise and resubmit your manuscript. Please attend to the reviewer's comments, which I believe may help to better strengthen the contributions of this manuscript to the field and public health practice. In addition, please attend to the following issues I noted in my review of the manuscript:

1. If citation 31 is published then please include a brief statement that reviews the RDS recruitment process (methods) and challenges reaching a representative sample using RDS (discussion). It is not enough to just point the reader to a subsequent manuscript. If this paper remains unpublished by the time of your re-submission, please provide sufficient detail on the RDS method and appropriate data on the representativeness of the sample recruited.

RESPONSE: We have now included the more extensive set of RDS diagnostics in Appendix 1 and more detail as to the representativeness of the survey to the discussion of study limitations, p.24.

2. Please indicate the variables missing data in Table 1 as done in Table 2 in the foot note

RESPONSE: We had added the numbers missing responses for each variable to Table 1. 

3. Please include the test statistics for comparison of recent HIV tests with and without missing date information in text. It is OK that these data are not presented in any of the tables.

RESPONSE: We have added the test statistics to the text, p.12.

I join the reviewers in their enthusiasm for the data you present on an important, but understudied population; highlighting both areas where some successes have been achieved and areas in need of targeted intervention efforts. Please provide a point by point response to each reviewers' response, and indicate in the response how it was addressed. This will help to facilitate a more timely review.

RESPONSE: Thank you, we respond to each comment and indicate where we have made changes to the manuscript.

Reviewer #1: This study uses RDS methodology to obtain a representative sample of MSM/TG in Johannesburg, in order to investigate: HIV prevalence, HIV testing behaviour in HIV negative individuals and viral suppression in HIV positive individuals. The study found a high prevalence of HIV, that the majority of negative MSM/TG had tested within the last 6 months, and poor viral suppression in PLHIV, although comparable to that of men in South Africa in general.

This is an important study adding to the evidence base attempting to understand HIV in this key population which has been historically neglected in Sub-Saharan Africa.

RESPONSE: Thank you.

I believe the manuscript could be improved by addressing the following comments:

1) Why was a cut-off of 200 used for viral suppression? The cut-off in South African treatment guidelines at the time of the study, as well as National Department of Health programme monitoring, is 400. The other commonly used cut-off at this time (2017) was 1000 (PEPFAR routine reporting; 2017 South African national HIV prevalence, incidence, behaviour and communication survey). In the new guidelines, 50 will be used. It would be more comparable to the other evidence and applicable to the South African setting to use 400 as a cut-off instead.

RESPONSE: In the absence of consistent viral load cut-points across studies and contexts, we chose to use 200 viral copies/ml as the cut-off, following CDC recommendations given in: 

Centers for Disease Control and Prevention. Questions and Answers for the General Public: Revised Recommendations for HIV Testing of Adults, Adolescents, and Pregnant Women in Health Care Settings. However, we agree with the reviewer that it would be helpful to present cascade findings using the South African guidelines. We think it makes most sense to use the cut-off of 50 viral copies/ml plasma to indicate viral suppression (our cut-off for detection was 40 copies/ml) in line with the 2019 South African guidelines, and because there is evidence that those with viral loads >50 copies/ml but less < 1000 copies/ml have poorer treatment outcomes compared to those with viral load <50 copies/ml (Hermans LE, Moorhouse M, Carmona S, Grobbee DE, Hofstra LM, Richman DD, et al. Effect of HIV-1 low-level viraemia during antiretroviral therapy on treatment outcomes in WHO-guided South African treatment programmes: a multicentre cohort study. Lancet Infect Dis. 2018;18(2):188-97).

We have also included in Appendix 2 the viral suppression n’s and RDS-weighted %’s for other cut-offs (200, 400 and 1000 copies/ml) used in the past and elsewhere in the literature, in order to aid comparison. 

2) In South Africa it is known that there is a large group of MSM who identify as straight (example Lane et al. High HIV Prevalence Among Men who have Sex with Men in Soweto South Africa: Results from the Soweto's Men's Study AIDS Behav (2011) 15:626–634). The RDS methodology does not appear to have penetrated into this group well (not unexpected). I think it is important to mention this in the discussion.

RESPONSE: We have added a note to the discussion of study limitations as follows, line 365: “Compared to another study of MSM in Soweto from 2008, we found a lower proportion of MSM who identified as heterosexual or ‘straight’(24), possibly indicating a difficulty in sampling this group.”

3) Although the viral suppression rates are similar to those of other men, the rate of viral suppression in men in South Africa is well below that of women. e.g. 2017 South African national HIV prevalence, incidence, behaviour and communication survey: viral suppression in men- 43%; viral suppression in women- 58%. Is it possible that the finding of similar viral suppression in MSM/TG is related to the fact that men in general experience such significant access barriers?

RESPONSE: Yes, we agree this is a possibility and we have now mentioned this in the discussion (see line 315): 

“It is plausible that in addition to challenges related to having sex with men, HIV-positive MSM/TG experience some of the same barriers to achieving viral suppression that men as whole do, as estimates for viral suppression among women are higher at than those of men overall (43% compare to 58% nationally among adults(40)).”

4) Reference 31: has this manuscript been accepted for publication yet? I do not believe it can be referenced as it stands. From the journal website:

"Do not cite the following sources in the reference list: Unavailable and unpublished work, including manuscripts that have been submitted but not yet accepted (e.g., “unpublished work,” “data not shown”). Instead, include those data as supplementary material or deposit the data in a publicly available database."

RESPONSE: We have now added more detailed RDS diagnostics to Appendix 1 instead of referencing this unpublished manuscript, and have added further detail to the Discussion.

5) There are a few minor editing comments:

- line 103: please change "18 years plus" to "18 years or older"

- Table 1, employment status, unemployed- there appears to be an error in the % column

RESPONSE: Thank you for bringing these issues to our attention. We have rectified them.

Reviewer #2: Review: HIV testing, care and viral suppression among men who have sex with men and transgender individuals in Johannesburg, South Africa

A very well written paper of major interest in the field. The authors estimated a very high HIV prevalence among MSM/TG in Johannesburg, and half of those HIV-positive were virally unsuppressed. These findings indicate gaps in the HIV care cascade that represent missed opportunities to improve the health of HIV-positive MSM/TG and to prevent ongoing HIV transmission.

I only have one major concern that requires the author’s revision before publication. The authors have conducted univariate and multivariate logistic regression to identify factors associated with HIV-testing in the previous 6 months amongst those HIV-negative; and to identify factors associated with viral suppression amongst those HIV-positive.

While the authors have correctly interpreted the odds ratios (OR). In this population, these binary outcomes (HIV testing, access to care and viral suppression) are common (>10%), it is often more desirable to estimate a relative risk (RR) or prevalence ratio since there is an increasing differential between the RR and odds ratio with increasing prevalence rates, and there is a tendency for some to interpret ORs as if they are RRs. The authors can estimate the RR in Stata with a log-binomial regression model or should there be convergence issues, apply the Poisson regression model with a robust error variance. A major consideration in public health research is need to communicate study results to a wider non-statistical community i.e. the medical community, general public, and policy makers. Poisson regression with robust variance and log-binomial regression provide correct estimates and are a better alternative for the analysis of this kind of study design with binary outcomes than logistic regression, as the relative risk or prevalence ratio is more interpretable and easier to communicate than the odds ratio.

Please see references below.

1. McNutt LA, Wu C, Xue X, Hafner JP. Estimating the Relative Risk in Cohort Studies and Clinical Trials of Common Outcomes. Am J Epidemiol 2003; 157(10):940-3.

2. Zou G. A Modified Poisson Regression Approach to Prospective Studies with Binary Data. Am J Epidemiol 2004; 159(7):702-6.

3. Sander Greenland , Model-based Estimation of Relative Risks and Other Epidemiologic Measures in Studies of Common Outcomes and in Case-Control Studies, American Journal of Epidemiology 2004;160:301-305

RESPONSE: Thank you for this suggestion. We have re-done our analyses using robust poisson regression to obtain weighted prevalence ratios instead of odds ratios for measure of association (still dropping seed participants and probability weighting using the inverse network weights in line with the RDS design). 

Interview Guides

We have added the survey instruments in English and Zulu as supplementary material (S3).

Data Access

Data from this study cannot be deposited publicly because of a high risk of deductive disclosure when reporting all of the information needed to validly analyse the data. This risk is made worse by the vulnerability of the population, which faces social stigma and has reported high levels of harassment, blackmail and abuse in other studies (eg Zahn 2016 (1)) and in our survey. 

The data for this study were collected using respondent driven sampling (RDS), a type of peer-referral or ‘snowball sampling’. Provided that assumptions are met and appropriate weighting is used, the data can be treated as a probability sample and used to obtain representative estimates of the population, eg HIV status. In practice, these assumptions are hard to meet and it is recommended to examine a set of RDS recruitment dynamics to assess the extent to which the sampling might deviate from assumptions, (see guidance on RDS diagnostics (2)). This requires information about participants’ social networks, information about their recruiter and those they recruit, and their position within sampling chains (eg whether one of only nine seed participants). We have given an overview description of the RDS recruitment as an appendix to this manuscript, and one that is specific to this study’s primary outcomes. We have taken care to do this while maintaining confidentiality, eg not giving HIV status of seed participants away via showing bottle net plots or recruitment trees by HIV status, but subsequent studies would need the full RDS information to assess the representativeness of the sample with respect to their specific study primary outcomes, eg to produce and examine convergence and bottleneck plots. We are concerned that there is a high risk of deductive disclosure to making the individual-level RDS data public, in combination with data about other characteristics of participants, including highly sensitive HIV and STI testing results and personal characteristics. Even if we removed individual pieces of identifiable information, we are not confident that this would truly anonymise the data. 

Although we cannot make study data publicly accessible at the time of publication, all authors commit to make the data underlying the findings of the study available, in compliance with the PLOS Data Availability Policy. Data can be requested via the London School of Hygiene and Tropical Medicine Research Operations Office Data Management Lead: alex.hollander@lshtm.ac.uk and the first author (elizabeth.fearon@lshtm.ac.uk) and Principal Investigator (tpalanee@wrhi.ac.za). Individuals requesting data should give their research objectives and indicate the list of requested variables, see S3 Survey Instrument. For data involving personally identifiable information or other sensitive data, data sharing is contingent on the data being handled appropriately by the data requester and in accordance with all applicable local requirements. 

(1) Zahn R, Grosso A, Scheibe A, Bekker LG, Ketende S, Dausab F, et al. Human Rights Violations among Men Who Have Sex with Men in Southern Africa: Comparisons between Legal Contexts. PLoS One. 2016;11(1):e0147156

(2) Gile KJ, Johnston LG, Salganik MJ. Diagnostics for Respondent-driven Sampling. J R Stat Soc Ser A Stat Soc. 2015;178(1):241-69

Added 26 March, following discussion with editor: for brevity, we propose the following data availability statement:

The data underlying this study cannot be made publicly accessible due to concerns about confidentiality when combining sensitive data about individuals and the respondent driven sampling (RDS) recruitment information needed to reflect the sampling design in analyses.  Although we cannot make study data publicly accessible at the time of publication, all authors commit to make the data underlying the findings of the study available` in compliance with the PLOS Data Availability Policy. Data can be requested via the London School of Hygiene and Tropical Medicine Research Operations Office Data Management Lead: alex.hollander@lshtm.ac.uk and the first author (elizabeth.fearon@lshtm.ac.uk) and Principal Investigator (tpalanee@wrhi.ac.za). Individuals requesting data should give their research objectives and indicate the list of requested variables, see S3 Survey Instrument. To protect the confidentiality of participants, data sharing is contingent on the data being handled appropriately by the data requester and in accordance with all applicable local requirements.

---

## [Decision Letter · Decision Letter 1]

27 May 2020

HIV testing, care and viral suppression among men who have sex with men and transgender individuals in Johannesburg, South Africa

PONE-D-19-20875R1

Dear Dr. Fearon,

We are pleased to inform you that your manuscript has been judged scientifically suitable for publication and will be formally accepted for publication once it complies with all outstanding technical requirements.

With kind regards,

Zixin Wang, PhD.

Academic Editor

PLOS ONE

Additional Editor Comments (optional):

Reviewers' comments:

Reviewer's Responses to Questions

**Comments to the Author**

1. If the authors have adequately addressed your comments raised in a previous round of review and you feel that this manuscript is now acceptable for publication, you may indicate that here to bypass the “Comments to the Author” section, enter your conflict of interest statement in the “Confidential to Editor” section, and submit your "Accept" recommendation.

Reviewer #1: All comments have been addressed

Reviewer #2: All comments have been addressed

2. Is the manuscript technically sound, and do the data support the conclusions?

Reviewer #1: (No Response)

Reviewer #2: Yes

3. Has the statistical analysis been performed appropriately and rigorously? 

Reviewer #1: (No Response)

Reviewer #2: Yes

4. Have the authors made all data underlying the findings in their manuscript fully available?

Reviewer #1: (No Response)

Reviewer #2: Yes

5. Is the manuscript presented in an intelligible fashion and written in standard English?

Reviewer #1: (No Response)

Reviewer #2: Yes

6. Review Comments to the Author

Reviewer #1: (No Response)

Reviewer #2: I have no additional comments or edits to add. The authors have attended to all concerns in a satisfactory way. Thank you.

7. PLOS authors have the option to publish the peer review history of their article (what does this mean?). If published, this will include your full peer review and any attached files.

Reviewer #1: No

Reviewer #2: Yes: Simbarashe Takuva

---

## [Editor Report · Acceptance letter]

4 Jun 2020

PONE-D-19-20875R1 

HIV testing, care and viral suppression among men who have sex with men and transgender individuals in Johannesburg, South Africa 

Dear Dr. Fearon:

I'm pleased to inform you that your manuscript has been deemed suitable for publication in PLOS ONE. Congratulations! Your manuscript is now with our production department. 

Kind regards, 

on behalf of

Professor Zixin Wang 

Academic Editor

PLOS ONE